# Plasma Metabolomics Study on the Impact of Different CRF Levels on MetS Risk Factors

**DOI:** 10.3390/metabo14080415

**Published:** 2024-07-27

**Authors:** Xiaoxiao Fei, Qiqi Huang, Jiashi Lin

**Affiliations:** College of physical Education, Jimei University, Xiamen 361021, China; fxiaoxiao1998@gmail.com (X.F.); qiqihuang072@gmail.com (Q.H.)

**Keywords:** blood metabolomics, cardiorespiratory fitness, metabolic syndrome risk factors, metabolic pathways

## Abstract

To investigate the metabolomic mechanisms by which changes in cardiorespiratory fitness (CRF) levels affect metabolic syndrome (MetS) risk factors and to provide a theoretical basis for the improvement of body metabolism via CRF in people with MetS risk factors, a comparative blood metabolomics study of individuals with varying levels of CRF and varying degrees of risk factors for MetS was conducted. Methods: Ninety subjects between the ages of 40 and 45 were enrolled, and they were categorized into low-MetS (LM ≤ two items) and high MetS (HM > three items) groups, as well as low- and high-CRF (LC, HC) and LCLM, LCLM, LCHM, and HCHM groups. Plasma was taken from the early morning abdominal venous blood. LC-MS was conducted using untargeted metabolomics technology, and the data were statistically and graphically evaluated using SPSS26.0 and R language. Results: (1) There were eight common differential metabolites in the HC vs. LC group as follows: methionine (↓), γ-aminobutyric acid (↑), 2-oxoglutatic acid (↑), arginine (↑), serine (↑), cis-aconitic acid (↑), glutamine (↓), and valine (↓); the HM vs. LM group are contrast. (2) In the HCHM vs. LCLM group, trends were observed in 2-oxoglutatic acid (↑), arginine (↑), serine (↑), cis-aconitic acid (↑), glutamine (↓), and valine (↓). (3) CRF and MetS risk factors jointly affect biological metabolic pathways such as arginine biosynthesis, TCA cycle, cysteine and methionine metabolism, glycine, serine, and threonine metabolism, arginine and proline metabolism, and alanine, aspartate, and glutamate metabolism. Conclusion: The eight common differential metabolites can serve as potential biomarkers for distinguishing individuals with different CRF levels and varying degrees of MetS risk factors. Increasing CRF levels may potentially mitigate MetS risk factors, as higher CRF levels are associated with reduced MetS risk.

## 1. Introduction

Metabolic syndrome (MetS) is characterized by a cluster of metabolic disorders, including hypertension, central obesity, dyslipidemia, and impaired glucose tolerance. Numerous studies have demonstrated that low cardiorespiratory fitness (CRF) is associated with an increased risk and mortality of MetS [1]. Both human and animal studies indicate that obesity, elevated fasting plasma glucose (FPG), and elevated insulin levels are common risk factors for MetS, particularly prevalent among individuals with low levels of CRF [2,3]. Further research is needed to elucidate the specific biochemical pathways influenced by high levels of CRF and their impact on MetS risk factors.

In recent years, there has been a growing emphasis among scholars on studying metabolic changes in the human body during exercise using metabolomics [4]. Methods such as nuclear magnetic resonance spectroscopy (NMR) and mass spectrometry (MS) enable the simultaneous measurement of a wide range of metabolites, revealing the biological remechanisms through which exercise improves health from a metabolic perspective, including its impact on MetS. Research over the past decade has primarily focused on comparing groups with significant disparities in CRF, such as those with the highest (>85%) and lowest (<15%) CRF levels or individuals with maximal oxygen consumption (VO_2max_) ranging from 38 to 48 mL^−1^kg^−1^min and above 60 mL^−1^ kg^−1^min [5,6]. This approach highlights that noticeable metabolic alterations are more evident when there are substantial differences in CRF levels. Current research has confirmed a link between CRF and several potential biomarkers related to MetS risk factors [7]. For instance, circulating levels of branched-chain amino acids (BCAAs) are significantly elevated in individuals affected by obesity, type 2 diabetes (T2D), and MetS compared to healthy controls [8]. For instance, there is evidence suggesting that circulating BCAA levels tend to decrease with increasing CRF levels or with improvements in health [9,10]. In a 16-week study, Sardeli et al. [11] combined aerobic and resistance exercise training for elderly women with MetS. Endurance training increased subjects’ CRF levels by 131%, while resistance training enhanced the leg press resistance load from 48% to 160%. Notably, the reevaluation of MetS risk factors post-exercise showed no significant changes, while metabolomic analysis revealed a significant increase in substrates involved in the tricarboxylic acid (TCA) cycle, including 2-ketobutyric acid, ketone, and acetoacetate. This metabolic response may account for the improvement in fatty acid oxidation due to exercise. Exercise therapies have the potential to considerably improve CRF levels in MetS patients, which, in turn, can improve their metabolic status. Consequently, assessing a person’s health only using their MetS risk factors may not be a reliable indicator of their general health.

This study employed High Performance Chemical Isotope Labeling (HP-CIL LC-MS) technology to conduct blood metabolomics analysis on populations with varying levels of CRF and MetS risk factors. The study aims to investigate the mechanisms through which CRF levels influence MetS risk factors by analyzing differences in blood metabolites among individuals with different CRF levels and degrees of MetS risk factors. Additionally, it seeks to provide insights into how enhancing CRF can potentially improve metabolic conditions in individuals with MetS risk factors, contributing to strategies for promoting health through exercise.

## 2. Materials and Methods

### 2.1. Study Subjects

A total of 100 participants aged 40–65 were recruited from Xiamen City Fujian Province, China. Inclusion in the study was based on meeting at least one of the following diagnostic criteria for MetS-related risk factors: (1)central obesity was defined as a waist circumference ≥ 90 cm for men and ≥85 cm for women; (2) fasting triglyceride (TG) level ≥ 1.7 mmol/L or under treatment; (3) high-density lipoprotein cholesterol (HDL-cholesterol) level ≤ 1.04 mmol/L or under treatment; (4) hypertension, defined as a systolic blood pressure (SBP) ≥ 130 mmHg or a diastolic blood pressure (DBP) ≤ 85 mmHg, or previously diagnosed and treated for hypertension; (5) hyperglycemia, defined as a fasting plasma glucose (FPG) level ≥ 6.1 mmol/L and/or 2-h postprandial glucose level ≤7.8 mmol/L, or previously diagnosed and treated for T2D. Subjects who did not complete the full trial or used vasoactive drugs (e.g., antihypertensive drugs, statins) or dietary supplements during the trial were excluded. The height, weight, waist circumference, and blood pressure of the subjects were measured (recorded to two decimal places). Fasting blood samples were collected after a 12-h fast for the measurement of FPG, TG, total cholesterol (TC), HDL-cholesterol, LDL-cholesterol, and homocysteine (Hcy). Measurements were performed using a Beckman Coulter automated biochemical analyzer (AU680, Beckman Coulter, Brea, CA, USA).

### 2.2. Cardiorespiratory Fitness Testing and Grouping

CRF was estimated using an indirect test to calculate the subjects VO_2max_. The step test method involved a 3-min exercise duration, during which subjects stepped up and down to a metronome set at 120 beats/min. Each subject wore a heart rate monitor to continuously record their heart rate throughout the experiment. The formula to estimate VO_2max_ is as follows: Step index = 100 × exercise time(s)/(2 × sum of heart rates in the 2nd, 3rd, and 4th 30-s intervals after stopping exercise).

For men: VO_2max_(mL kg^−1^ min^−1^) = 0.2588 × step index + 24.170.

For women: VO_2max_(mL kg^−1^min^−1^) = 0.1912 × step index + 17.264.

MetS grouping: Subjects with ≤2 MetS risk factors were assigned to the low-risk group (LM) and those with ≥3 MetS risk factors were assigned to the high-risk group (HM), based on the severity of MetS risk factors. There were 41 subjects in the LM group and 49 in the HM group.

CRF grouping: Of the subjects who took the CRF test, 11 were unable to finish it or participate and 79 participants provided valid data for the collection. Using tertiles, CRF was split into three levels as follows: low (LC); medium (MC); and high (HC). The LC group consisted of 26 individuals; the MC group of 26; and the HC group of 27.

CRF + MetS grouping: Based on CRF levels and MetS risk factors, subjects were divided into four groups as follows: low CRF and low MetS risk (LCLM), low CRF and high MetS risk (LCHM), high CRF and low MetS risk (HCLM), and high CRF and high MetS risk (HCHM). There were 18 subjects in the LCLM group, 8 in the LCHM group, 12 in the HCLM group, and 15 in the HCHM group.(see Figure 1)

### 2.3. Blood Sample Collection

Subjects fasted for 12 h and rested for 5 min to induce calmness. With their elbows elevated, 2 mL of venous blood were drawn into purple-top tubes containing anticoagulant. The tubes were gently inverted five to eight times, incubated at room temperature (20 to 25 °C) for 25–30 min, and then centrifuged at 3000 rpm for 10 min. The top layer of plasma was separated and stored in EP tubes pre-cooled to −80 °C for testing.

### 2.4. Metabolite Extraction

Plasma was divided into aliquots based on the requirements of the analysis. After thawing and vortexing, 30 µL of plasma from each sample was transferred into respective 1.5 mL centrifuge tubes. Each sample was divided into three parts as follows: one for single-channel analysis (30 µL per channel), one as a backup, and one for preparing mixed samples. For the mixed sample, 35 µL of plasma from each sample was pooled, carefully vortex-mixed to ensure uniformity, and labeled as a reference sample.

### 2.5. Protein Precipitation

To precipitate proteins, 90 µL of pre-cooled methanol (mass spectrometry-grade) was added to each centrifuge tube containing 30 µL of plasma. After thorough vortexing and centrifugation at low speed, the samples were placed in a −20 °C freezer for 1 h and subsequently centrifuged at high speed at 4 °C (12,000 rpm, 10 min). Ninety microliters of supernatant were transferred to new centrifuge tubes and dried under nitrogen.

### 2.6. Sample Labeling

For the secondary metabolomics analysis of amines and phenols, aliquoted samples were reconstituted by adding 25 µL of mass spectrometry-grade water. To each sample, 12.5 µL of buffer reagent A and 37.5 µL of ^12^C labeling reagent B (for single and mixed sample labeling) or ^13^C labeling reagent B (for mixed sample labeling only) were added. After vortex mixing, the samples were incubated at 40 °C for 45 min. To quench excess labeling reagent, 7.5 µL of reagent C was added post-incubation, followed by 30 µL of pH regulator D.

### 2.7. Sample Mixture

Following standard operating protocols, the labeled amine/phenolic secondary metabolome was analyzed using LC-UV. An aliquot of the ^13^C-labeled mixed sample was combined with the ^12^C-labeled individual sample for liquid-quality analysis, guided by the quantification results. The QC samples were prepared simultaneously with the liquid-quality analysis; specifically, equal volumes of mixed samples labeled with ^13^C and ^12^C were thoroughly mixed and used as QC samples. Each sample was prepared for analysis using liquid-quality analysis.

### 2.8. Analysis Condition and Data Quality Control and Metabolite Identification Results

LC-MS analysis was performed according to standardized operating procedures. Quality control and retention time calibration samples were analyzed every 15 injections to monitor instrument stability (see Table 1).

Quality Accuracy Check: As depicted in Figure 2a,b, in the amine/phenol channel analysis, the chromatographic peak with a mass-to-charge ratio (*m*/*z*) of 251.0849 was chosen as the reference peak for assessing the quality accuracy of data from 97 samples. Across all samples analyzed, the total signal intensity remained stable, and all scanned mass-to-charge ratios fell within the expected range, demonstrating the robust stability and accuracy of the collected data.

Retention Time Check: As depicted in Figure 2c, data from 7 retention time calibration standards were utilized to verify the retention times. All standard peaks were well-resolved, and the retention times of the standards across all calibration data were consistent, indicating the robust stability of the collected data’s retention times.

Metabolite Identification Results: Out of the detected metabolite peaks comprising 1851 features, 1679 peaks (90.7%) were accurately identified or inferred. As shown in Figure 2d, the three-tiered metabolite identification approach was applied as follows: Level 1, based on accurate molecular weight and retention time, identified 146 metabolites in the standard database (CIL Library); Level 2, utilizing the associated metabolite database (LI Library), identified an additional 214 metabolites; Level 3, where the remaining metabolites were matched in the MyCompoundID (MCID) database, resulting in matches for 347 metabolites in the 0-level reaction database, 657 metabolites in the 1-level reaction database, and 315 metabolites in the 2-level reaction database.

### 2.9. Statistical Analysis

The basic information of the subjects was statistically analyzed using SPSS 26.0 and GraphPad Prism 9.5 software. Results were expressed as mean ± SD, with statistically significant differences denoted by *p* < 0.05. The normality of the data was assessed using the Shapiro–Wilk test. The two independent samples *t*-test was applied when the data between two groups were normally distributed, and the non-parametric Mann–Whitney U-test was used for non-normal distribution. For comparisons involving three or more groups, one-way ANOVA was employed under the assumption of normal distribution and homogeneity of variances. Post hoc analysis was conducted using the LSD method for multiple comparisons, while the Kruskal–Wallis H-test was used for non-normally distributed data or when variances were not uniform.

Blood samples were analyzed using LC-MS metabolomics technology, and the data were processed and analyzed with Iso MS Pro 1.2.16 software. A three-tiered metabolite identification approach was used to structurally identify compounds and complete the initial metabolite data. The normalized data matrix was imported into SIMCA-*p* 14.1 software for multivariate data analysis, using partial least squares discriminant analysis (PLSA-DA) and orthogonal partial least squares-discrimination analysis (OPLS-DA) to compare differences in plasma metabolome. Regarding the variable importance projection (VIP) > 1.0 and nonparametric tests *p* < 0.05 as standard, choose differential metabolites. To prevent the overfitting of the OPLS-DA model, response ordering tests (200 random permutations) and cross-validation are used to evaluate the model’s quality.

Following data processing, the fold change (FC) values for each group were determined using the sample mean ratio. (*p*-values from *t*-tests, *q*-values from FDR-adjusted *p*-values.) Differential metabolites between groups were selected based on FC values, *p*-values, and *q*-values obtained by volcano plot analysis, Venn diagram analysis, and VIP values. Metabolic pathway analysis was performed on the MetaboAnalyst 5.0 metabolomics analysis platform (https://www.metaboanalyst.ca/, accessed on 7 August 2022), and metabolic pathways were identified using the Kyoto Encyclopedia of Genes and Genomes (KEGG).

## 3. Results

### 3.1. Subjects

Subjects were classified by VO_2max_ into LC (28.90 ± 1.37, *n* = 26), MC (35.78 ± 3.41, *n* = 26), and HC (43.18 ± 1.46, *n* = 27) (*p* < 0.05). Compared to the LC group, the MC group had a significantly lower age (*p* < 0.05), while the HC group had significant differences in height, weight, WC, and HDL-cholesterol (*p* < 0.05). Compared to the MC group, HDL-cholesterol was significantly lower in the HC group (*p* < 0.05).

Subjects were divided into LM and HM groups based on the degree of MetS risk factors. Compared to the LM group, significant differences were observed in VO_2max_, height, weight, BMI, WC, FPG, TG, TC, HDL-C, and Hcy in the HM group (*p* < 0.05) (see Table 2).

A total of 53 subjects were included based on the CRF level and degree of MetS risk factors. Compared to the LCLM group, significant differences were observed in weight, BMI, WC, and HDL-cholesterol in the LCHM group (*p* < 0.05), in VO_2max_, height, weight, and WC in the HCLM group (*p* < 0.05), and in VO_2max_, height, weight, BMI, WC, SBP, TG, HDL-cholesterol, and Hcy in the HCHM group (*p* < 0.05). Compared to the LCHM group, significant differences were observed in VO_2max_ and TG in the HCLM group (*p* < 0.05) and in VO_2max_ and HDL-cholesterol in the HCHM group (*p* < 0.05). Compared to the HCLM group, significant differences were observed in WC, SBP, FPG, TC, and HDL-cholesterol in the HCHM group (see Table 3).

### 3.2. PLS-DA

Figure 3a shows the PLS-DA score plot of plasma metabolites for subjects with different CRF levels. Studies show that there is overlap between the plasma samples from the MC and HC groups, as well as between the LC and MC groups, making it impossible to identify between the three subject groups’ plasma samples.

The PLS-DA score plot of plasma metabolites for subjects with varying degrees of MetS risk factors is displayed in Figure 3b. The findings suggest that although there is a trend of separation, the plasma samples from the LM and HM groups cannot be fully differentiated from one another.

The PLS-DA score plots for the comparisons of the LCLM vs. HCLM, LCHM vs. HCHM, LCLM vs. HCHM, and LCHM vs. HCLM groups are shown in Figure 3c–e and Figure 3g,h, respectively. The findings demonstrate the total plasma sample separation, with no overlap, among these five comparison groups. The comparison between the HCLM and HCHM groups is depicted in Figure 3f, where the PLS-DA score plot indicates an imperfect separation of plasma samples between these groups with overlap.

The cross-validation results for each group are as follows: as shown in Table 4, the models of LC vs.MC vs. HC groups have relatively small values of R^2^X, R^2^Y, and Q^2^, indicating that this model lacks reliability and predictability. Q^2^ < 0.5 (HCLM vs. HCHM and LCHM vs. HCLM), indicating that the model is valid but lacks predictability. Models of LM vs. HM, LCLM vs. HCLM, LCHM vs. HCHM, LCLM vs. LCHM, and LCLM vs. HCHM groups demonstrate reliability and predictability.

R^2^X denotes the explanatory rate of the model to the X matrix; R^2^Y denotes the explanatory rate of the model to the Y matrix; and Q^2^ denotes the predictive power of the model. The closer the three models are to 1, the better, and Q^2^ > 0.5 is accepted.

### 3.3. OPLS-DA

Filtering out the irrelevant orthogonal signals by OPLS-DA analysis made the differences between groups more obvious and the differential metabolites obtained more reliable. OPLS-DA was used to analyze each differential comparison group of the CRF, MS, and CRF + MS groups and to verify whether the model was overfitted with cross-validation and response ranking

The metabolic status of the subjects in both groups was determined by the complete separation of sample points in the LC vs. HC group (see Figure 4A,a); the results show that the model is valid and there is no overfitting (R^2^X = 0.259, R^2^Y = 0.974, Q^2^ = 0.607). The separation trend in plasma sample points was less distinct in both the MC vs. HC and LC vs. MC groups when compared to the LC vs. HC group(see Figure 4B,C,b,c). This suggests a less pronounced metabolic status in these groups (R^2^X = 0.162, R^2^Y = 0.930, Q^2^ = 0.310; R^2^X = 0.139, R^2^Y = 0.810, Q^2^ = −0.152); the models were not overfitted, as demonstrated by the response order tests, but the cross-validation results revealed that Q^2^ < 0.5 for the two comparison groups, indicating that the models’ predictability was lacking. The MS group’s plasma isolation sample points in the LM vs. HM group were completely separated, and the model (R^2^X = 0.415, R^2^Y = 0.998, Q^2^ = 0.520) was observable (see Figure 4D,d). The model’s validity and predictability are all demonstrated by the cross-validation results, and these qualities are further demonstrated by the outcomes of the matching ranking test.

The OPLS-DA scores for each group in the CRF + MS group were as follows(see Table 5): (1) The sample points of the four groups—LCLM vs. HCLM, LCHM vs. HCHM, LCLM vs. HCHM, and LCLM vs. HCHM—were completely segregated, indicating significant differences in the participants’ metabolic states (see Figure 4E−G,I,e–g,i). The results of the 200-response sequencing test confirmed the validity and predictability of the models in all four groups, and the cross-validation results showed their reliability (R^2^X = 0.296, R^2^Y = 0.994, Q^2^ = 0.548; R^2^X = 0.262, R^2^Y = 0.979, Q^2^ = 0.520; R^2^X = 0.282, R^2^Y = 0.966, Q^2^ = 0.613; R^2^X = 0.342, R^2^Y = 0.997, Q^2^ = 0.651). (2) The HCLM vs. HCHM group and the LCHM vs. LCHM group have completely separated sample points, and the metabolic differences between the subjects are obvious(see Figure 4H,J,h,j), which makes the model valid and reliable, but less predictive (R^2^X = 0.321, R^2^Y = 0.989, Q^2^ = 0.028; R^2^X = 0.192, R^2^Y = 0.958, Q^2^ = 0.459).

The metabolism analysis demonstrated that each group’s samples were well-clustered, while there was a significant separation between the groups. R^2^Y and Q^2^ values were computed in the OPLS-DA modeling study to demonstrate the model’s validity. Q^2^ values were also used to demonstrate the model’s validity. A 200 random permutation test was conducted to demonstrate that the model did not overfit (Q^2^ < 0) and to validate its reliability.

### 3.4. Volcano Plot Analysis

Given the results of the OPLS-DA analysis, volcano plot analyses were performed for the LC group vs. the HC group, the LM group vs. the HM group, the LCLM group vs. the HCLM group, the LCHM group vs. the HCHM group, the LCLM group vs. the LCHM group, and the LCLM group vs. the HCHM group. As can be seen in Figure 5a, 83 metabolites were upregulated and 16 metabolites were downregulated in the HC vs. LC group comparison; in Figure 5b, 84 metabolites were upregulated and 78 metabolites were downregulated in the HM vs. LM group comparison; in Figure 5c, 40 metabolites were upregulated and 11 metabolites were downregulated in the HCLM vs. LCLM group comparison; in Figure 5d, 121 metabolites were upregulated and 106 metabolites were downregulated in the HCHM vs. LCHM group comparison; in Figure 5e, 112 metabolites were upregulated and 99 metabolites were downregulated in the LCHM vs. LCLM group comparison; and in Figure 5f, 140 metabolites were upregulated and 61 metabolites were downregulated in the HCHM vs. LCLM group comparison.

### 3.5. Venn Diagram Analysis

Eight common differentially expressed metabolites were identified after excluding metabolites that could not be matched or identified. These eight frequently occurring differentially expressed metabolites may serve as biomarkers for distinguishing different levels of MetS risk factors and CRF levels. Methionine, γ-aminobutyric acid, 2-oxoglutaric acid, arginine, serine, cis-aconitic acid, glutamine, and valine showed significant differences in both comparison groups.

In the comparisons, 2-oxoglutaric acid and cis-aconitic acid were screened in the HCLM vs. LCLM group; methionine, serine, 2-oxoglutaric acid, arginine, and glutamine in the HCHM vs. LCHM group; and glutamine, methionine, and arginine in the LCHM vs. LCLM group. Additionally, 2-oxoglutaric acid, arginine, serine, cis-aconitic acid, glutamine, and valine were screened in the HCHM vs. LCLM group (see Figure 6). 

The trends in the common differential metabolites in the comparison groups were different, as shown in Figure 6 and Table 6: (1) methionine: HC vs. LC and HCHM vs. LCHM (↓), HM vs. LM and LCHM vs. LCHM (↑); (2) γ-aminobutyric acid: HC vs. LC (↑), HM vs. LM (↓); (3) 2-oxoglutaric acid: HC vs. LC (↑), HM vs. LM (↓); HCLM vs. LCLM, HCHM vs. LCHM, and HCHM vs. LCLM (↑) (4) arginine: HC vs. LC (↑), HM vs. LM (↓), HCHM vs. LCHM, HCHM vs. LCHM (↑), LCHM vs. LCLM (↓) (5) serine: HC vs. LC (↑), HM vs. LM (↓), HCHM vs. LCLM (↑), LCHM vs. LCHM (↓); (6) cis-aconitic acid: HC vs. LC (↑), HM vs. LM (↓), HCLM vs. LCLM and HCHM vs. LCLM (↑); (7) glutamine: HC vs. LC (↓), HM vs. LM (↑), HCHM VS.LCHM and HCHM vs. LCLM (↓); (8) Valine: HC vs. LC (↓), HM vs. LM (↑); HCHM vs. LCHM and HCHM vs. LCLM (↓).

Six common differentially expressed metabolites were identified across three comparison groups as follows: HM vs. LM, HC vs. LC, and HCHM vs. LCLM. As illustrated in Figure 7a–d, the following were found to be upregulated in the HC vs. LC and HCHM vs. LCLM group: 2-oxoglutaric acid, arginine, serine, and cis-aconitic acid; however, these same variables were found to be downregulated in the HM vs. LM group; similarly, Figure 7e,f revealed that valine and glutamine were both downregulated in the HCHM vs. the LCLM group and the HC vs. the LC group, but upregulated in the HM vs. the LCLM group.

### 3.6. Metabolite Pathway Analysis

For the purpose of determining the primary metabolic pathways impacting each group’s metabolic status, Metaboanalyst 5.0 was also used for the topological analysis and enrichment of metabolic pathways of differential metabolites in each comparison group.

As illustrated in Figure 8a,b, the metabolic pathways of arginine biosynthesis, the TCA cycle, cysteine and methionine metabolism, glycine, serine, and threonine metabolism, arginine and proline metabolism, and alanine, aspartate, and glutamate metabolism were the metabolic pathways that were jointly affected by the HC vs. LC group and the HM vs. LM group.

The metabolic pathways of arginine biosynthesis, arginine and proline metabolism, as well as alanine, aspartate, and glutamate metabolism were jointly affected by the HC vs. LCLM group, the HCHM vs. LCHM group, the LCHM vs. LCLM group, and the HCHM vs. LCLM group, as shown in Figure 8c–f.

## 4. Discussion

This study aimed to investigate how variations in CRF levels impact MetS risk factors by conducting a metabolomics analysis on individuals with varying degrees of MetS. The study’s primary conclusions were as follows: (1) By comparing intergroup differences between the HM vs. LM and the LC vs. HC, we identified eight common differential metabolites that serve as potential biomarkers for distinguishing between different CRF levels and degrees of MetS risk factors as follows: methionine, γ-aminobutyric acid, 2-oxoglutaric acid, arginine, serine, cis-aconitic acid, glutamine, and valine. (2) Six common differential metabolites, (methionine, γ-aminobutyric acid, 2-oxoglutaric acid, arginine, serine, cis-aconitic acid, glutamine, and valine) were found after the additional comparative analyses of the results of differential metabolites screened in the HCHM vs. LCLM, HC vs. LC, and HM vs. LM. These metabolites primarily affect arginine biosynthesis, the tricarboxylic acid (TCA) cycle, cysteine and methionine metabolism, glycine, serine and threonine metabolism, arginine and proline metabolism, and alanine, aspartate, and glutamate metabolism, among other biometabolic pathways.

The study identified how CRF levels influence metabolic status across varying degrees of MetS through metabolomic analysis, shedding light on CRF’s impact on MetS risk via specific metabolic pathways. Changes in these metabolites may correlate with diverse CRF levels and MetS risk, offering novel perspectives for clinical diagnosis and intervention. The findings indicate that higher CRF levels can alleviate the detrimental effects of MetS risk factors, suggesting that enhancing fitness could effectively prevent or treat MetS. Moreover, the identification of six common differential metabolites impacting multiple vital biometabolic pathways enhances our comprehension of metabolic network complexity and proposes potential targets for intervention.

### 4.1. Differences in Plasma Metabolites between CRF Levels and Different Degrees of Mets Risk Factors

There are eight common differentially expressed metabolites in the HC vs. LC and HM vs. LM groups, namely methionine, γ-aminobutyric acid, 2-oxoglutaric acid, arginine, serine, aconitic acid, glutamine, and valine. They are potential biomarkers for distinguishing different CRF levels and degrees of MetS risk factors. Methionine is one of the essential amino acids in the human body and serves as an important cellular antioxidant [12]. It also participates in the regulation of insulin sensitivity. The present study observed a tendency for methionine downregulation in individuals with high CRF levels when compared to those with low CRF levels. Conversely, there was a tendency for methionine upregulation in subjects with high MetS risk factors compared to those with low MetS risk factors. Because exercise-induced oxidative stress inhibits methionine, both long-term endurance exercise can result in a substantial drop in plasma methionine levels [13]. Oxidative stress (OS) results from an imbalance between reactive oxygen species (ROS) production and antioxidant mechanisms. Regular moderate-intensity exercise has been demonstrated to enhance antioxidant defense mechanisms, thereby reducing mitochondrial oxidative damage [14,15]. However, the specific mechanism underlying the relationship between methionine depletion and its impact on oxidative stress induced by endurance exercise remains unclear. Past studies have demonstrated a positive association between blood levels of branched-chain and aromatic amino acids and MetS risk. However, the association of other essential amino acids such as methionine, threonine, and lysine with MetS remains poorly understood [16]. Azab et al. [17] discovered a favorable correlation between the prevalence of MetS and plasma levels of Met, Threonine, and Lysine. γ-aminobutyric acid (GABA) is generated from glutamate through the catalysis of glutamate decarboxylase. In this study, we observed significantly higher plasma levels of GABA in subjects with high CRF levels compared to those with low CRF levels. Conversely, plasma levels of GABA were significantly lower in subjects with high MetS risk factors compared to those with low MS risk factors. Yan et al. [18] found that, among male college students aged 18 to 29, serum GABA levels were significantly higher in groups with regular endurance exercise and strength exercise habits compared to those without such habits. Additionally, the GABA content of the endurance exercise habit group was significantly higher than that of the strength exercise habit group. These findings imply that prolonged endurance training will raise the level of GABA in the blood. Moreover, GABA modulated high-fat-induced obesity, glucose intolerance, and insulin resistance (IR). Oral GABA administration was shown to prevent weight gain, significantly lower FPG levels, improve glucose tolerance, and subsequently increase insulin sensitivity in high-fat-induced mice. Additionally, GABA significantly increased the number of regulatory T cells, which can control the development of obesity-induced IR and T2D [19]. Therefore, increasing CRF levels bringing about the upregulation of GABA content may be very favorable for intervening obesity and T2D occurrence.

Both Met and GABA can serve as potential biomarkers to distinguish CRF levels and the degree of MetS risk factors. Whether they can explain the mechanism by which high CRF levels improve the metabolism of individuals with MetS risk factors still needs further investigation.

### 4.2. Higher CRF Levels Reduce the Risk of MetS Risk Factors

In order to investigate the metabolomics mechanisms by which high levels of CRF affect high MetS risk factors, the interaction between CRF and MS revealed that high CRF levels, acting as a protective factor against high MetS risk, correlated with six metabolites. Four metabolites exhibited a downregulation trend in HM and an upregulation trend in HCHM. Conversely, two metabolites showed an upregulation in the HM and a downregulation in HCHM. The above metabolite trends indicate that the metabolite levels in the high MetS risk factor group were reversed due to the effect of high levels of CRF, which suggests that high levels of CRF can mitigate the detrimental effects of high MetS risk factors and contribute to improved metabolic status compared to individuals with low MetS risk factors.

α-Ketoglutaric acid is another name for ketoglutaric acid (Akg), an intermediary of the TCA cycle that is essential to several metabolic activities. We found a trend towards the upregulation of 2-oxoglutarate in the plasma of subjects with high levels of CRF compared to subjects with low levels of CRF, and a trend towards the downregulation of 2-oxoglutarate in subjects with high MS risk factors compared to subjects with low MS risk factors. Yuan et al. [20] examined peripheral blood samples from mice that were either inactive control mice or mice that had undergone acute resistance exercise (40 min of ladder climbing with a 10% body weight load). It was discovered that Akg concentrations were negatively correlated with HbA1c and several metabolic risk factors, including BMI, WC, adiposity, and body weight [21,22]. This helps to explain why patients with diabetes mellitus (DM) had significantly lower levels of Akg than healthy controls. But, when the HCHM and LCLM groups were compared, the trend in Akg increased rather than decreased, even though the HCHM group had a much higher weight, BMI, and WC than the LCLM group. The reason for this phenomenon could be that those in the HCHM group who have high CRF levels have already somewhat changed the metabolic status of people who have high MetS risk factors. Leibowitz et al. [23] found that exercise caused a significant increase in the amount of Akg in the blood, and that 2-oxoglutarate is produced from glutamine and glutamate through deamidation, ultimately entering the TCA cycle. Exercise also reduces levels of glutamate and leucine, whose breakdown can further elevate 2-oxoglutarate levels. Furthermore, elevated levels of glutamate are observed in populations with MetS when compared to those without. This supports the idea that exercise boosts CRF, promotes the synthesis of 2-oxoglutarate in the TCA cycle, and helps alleviate the adverse effects of MetS risk factors.

Arginine (Arg) is an amino acid involved in various metabolic pathways. It serves as a substrate for the NOS enzyme family, which produces nitric oxide (NO), a key molecule involved in normal endothelial function, insulin sensitivity, and metabolic profile. Research has demonstrated that during cardiopulmonary exercise testing, the concentration of arginine rises noticeably in heart failure patients, and, in those with greater exercise capacity, the concentration of arginine is positively connected to exercise capacity [24]. This is consistent with the views expressed in this essay. Additionally, Holz et al. [25] showed that 30 min of exercise at a 75% VO_2max_ intensity resulted in differential levels of glutamate and arginine. This suggests that exercise training positively affects endothelial function by significantly enhancing the endothelium-dependent dilatation of the musculoskeletal system, leading to notable changes in arginine levels, which are linked to NO production. Arg supplementation has been shown to considerably improve endothelial [26,27], β-cell [28], and oxidative stress function [29] in patients with T2D, according to current studies. Consistent with Palmnas’ findings, this study concluded that the level of arginine in the HCHM group was significantly higher compared to that in the LCLM group [30]. In conclusion, there is evidence supporting the idea that physical activity can enhance health by increasing arginine levels in the body and, to a lesser extent, reducing the severity of illnesses. Exercise could potentially serve as an alternative to arginine supplementation therapy; however, additional research is necessary to confirm this hypothesis.

Serine (Ser) is a neuronal nutrition factor and a precursor of numerous important molecules, including D-serine, phosphatidylserine, sphingomyelin, and glycine. Low levels of Ser have been associated with risk factors such as metabolic syndrome, DM, obesity, and IR. In this study, Ser levels were observed to be higher in subjects with high CRF levels and lower in subjects with high MetS risk factors. Studies have shown that low levels of Ser, glycine, and threonine can differentiate between individuals with poor metabolic profiles (≥2 MetS risk factors) and those with good metabolic profiles (<2 MetS risk factors) [31]. In order to measure energy expenditure and physical activity levels, Lee et al. [13] compared the highest (50%) and lowest (50%) participants in both groups using the doubly labeled water. Ser levels were significantly higher in subjects with HCHM when compared to LCLM. This change may be related to the potential mechanism of the negative correlation between sphingolipid metabolism and Ser. Dube et al. [32] found that both diet-induced weight loss and exercise training can improve IR, but only exercise can reduce the levels of ceramide. Ser may be utilized in obese conditions as a precursor of sphingolipids, potentially leading to ceramide accumulation in insulin-sensitive organs (such as the liver and muscle), thereby accelerating the onset of IR [33]. High levels of exercise with high CRF may downregulate Serine Palmitoyl Transferase (SPT), the enzyme catalyzing the initial step of de novo sphingolipid synthesis by utilizing Ser and palmitoyl coenzyme A to generate 3-ketosphinganine. The lipid-mediated suppression of insulin signaling, and ceramide buildup are both prevented by the inhibition of SPT. Thus, it makes sense to speculate that SPT at high CRF levels would be crucial for ceramide metabolism and IR in individuals.

Cis-aconitic acid, along with its derivatives, is involved in the metabolism of the tricarboxylic acid (TCA) cycle, acetic acid, and dicarboxylic acids. In the TCA cycle, cis-aconitic acid catalyzes the conversion of citrate to isocitrate, thereby playing a crucial role in the completion of the entire TCA cycle process. In this study, we observed that the concentration of cis-aconitic acid was significantly higher in the high-level CRF group when compared to the low-level CRF group. Shi et al. [34] took blood samples from 20 marathon runners both before and after the race. They discovered that the levels of cis-aconitic acid, pyruvic acid, malic acid, glyceric acid, and galacturonic acid were much higher after the race than they were before. This suggests that cis-aconitic acid levels in the body can be considerably raised by aerobic endurance activity. The HM group exhibited significantly lower levels of cis-aconitic acid compared to the LM group. Additionally, Zou et al. [35] observed that women with T2D and severe obesity had lower concentrations of TCA cycle intermediates, including citric acid, cis-aconitic acid, and γ -ketoglutaric acid. These findings suggest that severe obesity and T2D may be associated with an impaired TCA cycle. In this study, the HCHM group exhibited significantly higher levels of cis-aconitic acid compared to the LCLM group. This may be attributed to the fact that the HCHM group engaged in long-term aerobic endurance exercise, where energy is predominantly derived from the aerobic metabolism of fats and carbohydrates. This enhances the activity of the body’s TCA cycle and leads to the accumulation of various TCA cycle intermediates, resulting in significant increases in metabolites associated with the metabolism pathways of acetaldehyde and dicarboxylic acid. To explain this phenomenon, individuals with high CRF levels exhibited a greater capacity for the TCA cycle when compared to those with low CRF levels and low MetS risk factors, despite individuals with high MetS risk factors showing lower levels of cis-aconitic acid.

Glutamine constitutes 20% of the total pool of free amino acids in the human body, making it the most abundant amino acid [36]. The liver, lungs, and adipose tissue are the primary sources of glutamine in the human blood as they are capable of synthesizing and releasing it. According to this study, the HC group’s glutamine level was substantially lower than the LC group. Several research works have demonstrated that prolonged endurance training or high-intensity training causes blood glutamine levels to drop [37].Although the mechanisms underlying this phenomenon are unknown, they could be due to an increase in glutamine absorption by other tissues, a decrease in muscle-released glutamine, or a decrease in muscle-synthesized glutamine [38]. However, glutamine levels were significantly higher in the HM group than in the LM group, where D-glutamine and D-glutamate metabolic pathways were significantly affected. Alsoud et al. [39] found that the glutamine level in the pre-DM MetS group was 4.8 times higher than that of the control group, and in the normoglycemic MetS group, it was 3.5 times higher. Notably, glutamine can be converted to glutamate, which synthesizes glutathione along with cysteine and glycine, so dysregulated glutathione synthesis may exacerbate the pathogenesis of DM [40]. Glutamate levels in the HCHM group were lower than in the LCLM group. Additionally, Lee et al. [13] observed that after a 12-week intervention combining strength and endurance exercises, concentrations of several metabolites involved in glutathione biosynthesis—including glutamate, total cysteine, total glutathione, creatinine, and taurine—were reduced. This suggests that chronic exercise may decrease glutathione biosynthesis. This effect could be attributed to enhanced insulin sensitivity and improved mitochondrial function, particularly at high levels of CRF.

Valine is one of the BCAAs, which is mostly metabolized in skeletal muscle. Plasma levels of valine were lower in the HC group compared to in the LC group; plasma levels of valine were higher in the HM group compared to in the LM group. Based on recent research, BCAA may be a biomarker for obesity, IR, and T2D. People with greater valine levels had a 36% higher chance of developing T2D compared to those with lower levels [41]. Wientzek et al. [18] found that increasing high-intensity physical activity (>6MET/h) by 1 h per day could reduce BCAA levels by 185% of the standard deviation, possibly due to its association with the TCA cycle. Valine undergoes oxidation and decomposition within the mitochondria of skeletal muscles, generating succinyl coenzyme A to enter the TCA cycle. Given the close relationship between CRF, mitochondrial density, and oxidative enzyme activity in skeletal muscles, it can be inferred that the active TCA cycle promotes the breakdown of valine, thereby reducing its concentration in the blood. This study also found that, when compared to the LCLM group, valine levels decreased in the HCHM group. Studies have demonstrated that aerobic exercise reduces inflammation, enhances BCAA aminotransferase activity, and upregulates the expression of genes related to the TCA cycle [42,43]. These alterations in enzyme activity encourage BCAA breakdown, which lowers plasma levels of the amino acid.

In summary, 2-oxoglutaric acid, arginine, serine, cis-aconitic acid, glutamine, and valineine are not only potential biomarkers reflecting different levels of CRF and different degrees of MetS risk factors, but also demonstrate that high CRF levels are protective factors influencing changes in the metabolic status of MetS risk factors.

### 4.3. Limitation

The metabolomic analysis of blood from MetS risk factor populations with varying CRF levels was conducted using LC-MS technology to compare metabolic profiles across varying CRF levels and to assess the impact of CRF on different stages of MetS risk factors, elucidating the metabolic mechanisms underlying how changes in CRF levels influence MetS risk factors, challenging existing paradigms regarding the impact of CRF on MetS risk factors. Nevertheless, further investigation is warranted to address the following unresolved issues. The blood metabolic profiles of populations at risk for MetS with varying levels of CRF were comprehensively analyzed using untargeted metabolomics technology to elucidate the impact of CRF on factors related to MetS; future studies could include quantitative analyses using targeted metabolomics technology to further investigate the diagnostic relevance of identified biomarkers. This study exclusively utilized metabolomics techniques due to experimental constraints. Exercise metabolomics research aims to integrate various omics approaches (e.g., genomics, proteomics) to comprehensively elucidate the biological implications of exercise. Therefore, the integration of other omics techniques to provide a comprehensive understanding of the biological effects of exercise.

## 5. Conclusions

Untargeted metabolomics technology was used to determine the plasma metabolic profiles of MetS risk factor populations with different CRF levels. Based on the comprehensive screening and validation procedures, eight metabolites were identified as markers distinguishing MetS risk factor populations based on CRF levels, namely a-ketoglutarate, cis-aconitic acid, arginine, γ-aminobutyric acid, serine, valine, methionine, and glutamine, indicating significant metabolic profile differences between MetS risk factor populations with high and low CRF levels. Arginine biosynthesis, the TCA cycle, and cysteine and methionine metabolism were markedly influenced by high CRF levels, implying a potential role of high CRF levels in metabolic improvements among individuals with MetS risk factors.

## Figures and Tables

**Figure 1 metabolites-14-00415-f001:**
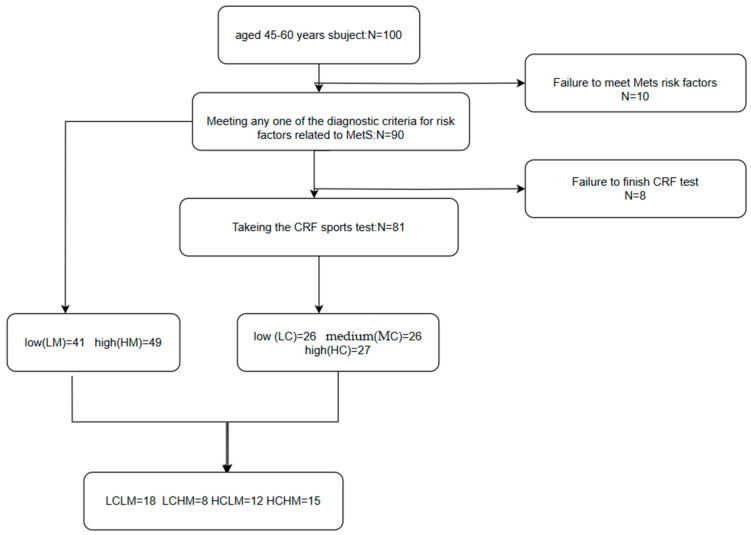
Flowchart of different metabolic syndrome MetS and cardiorespiratory fitness CRF groupings. Participants were categorized into the low-risk group and the high-risk group (HM) if they had ≥3 MetS-related risk factors; CRF was categorized into low CRF level (LC), medium CRF level (MC), and high CRF level (HC) using the 3-quartile method; participants were classified into four groups based on the CRF level and the number of MetS-related risk factors: low CRF and low MS risk factor group (LCLM), low CRF and high MS risk factor group (LCHM), high CRF and low MS risk factor group (HCLM), and high CRF and high MS risk factor group (HCHM).

**Figure 2 metabolites-14-00415-f002:**
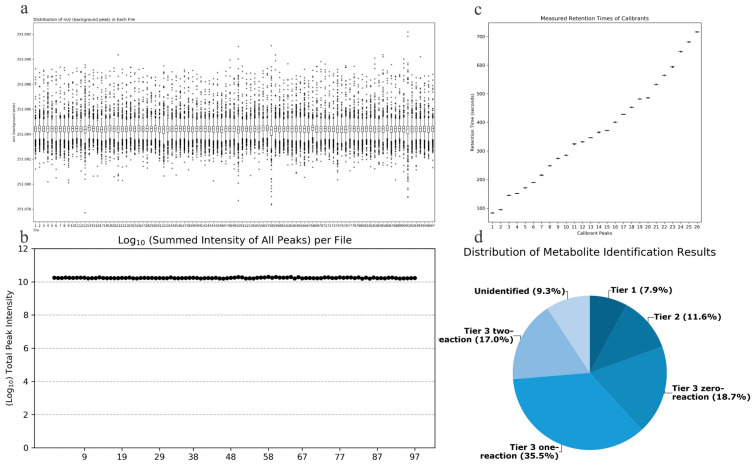
(**a**) Total signal intensity in the amine/phenol-based channel sub-sample analysis; (**b**) Background peak mass distribution in the amine/phenol-based channel sub-sample analysis; (**c**) Retention time assay results; (**d**) Distribution of identified metabolites in different tiers of the database.

**Figure 3 metabolites-14-00415-f003:**
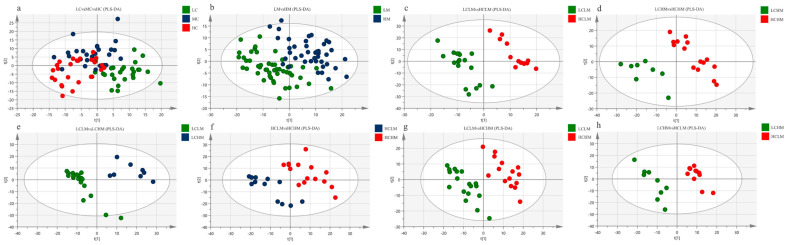
PLS-DA Score Plot: (**a**) LC vs. MC vs. HC group; (**b**) LM vs. HM group (**c**) LCLM vs. HCLM; (**d**) LCHM vs. HCHM; (**e**) LCLM vs. LCHM; (**f**) HCLM vs. HCHM; (**g**) LCLM vs. HCHM; (**h**) LCHM vs. HCLM.

**Figure 4 metabolites-14-00415-f004:**
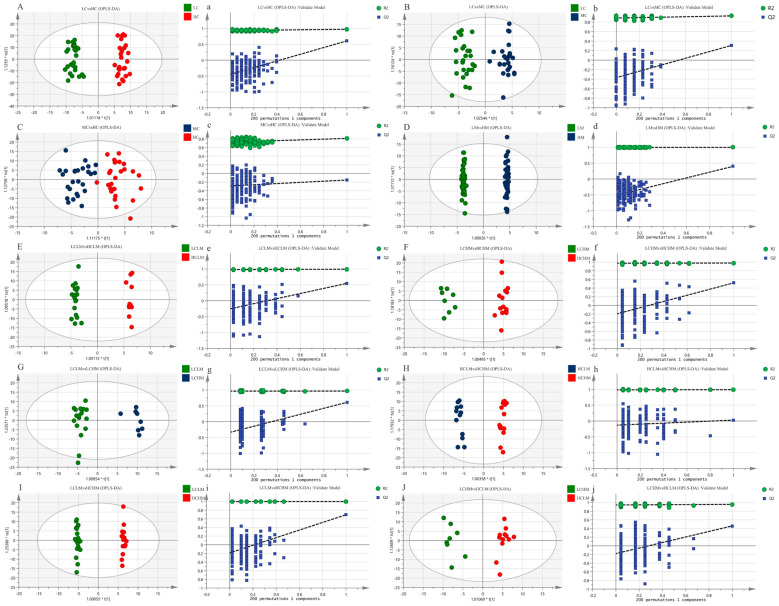
The OPLS-DA score plots (**A**–**J**) depict the scores for each group, while the response ranking test plots (**a**–**j**) display the results of the response ranking test. In the ranking test, the horizontal axis represents the correlation between the *Y* values of the random grouping and the *Y* values of the original grouping, while the vertical axis represents the R^2^ and Q^2^ scores.

**Figure 5 metabolites-14-00415-f005:**
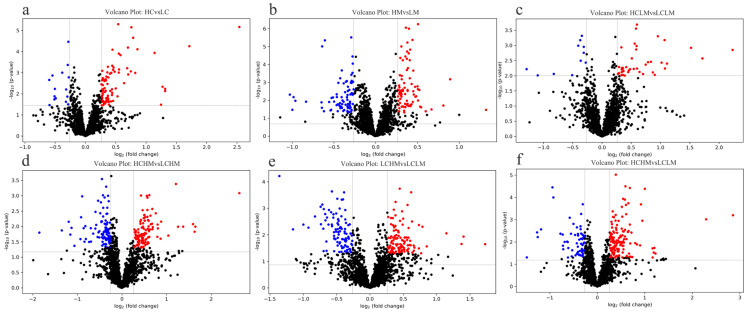
Differential metabolite volcano plot (**a**) HC vs. LC, (**b**) HM vs. LM, (**c**) HCLM vs. LCLM, (**d**) HCHM vs. LCHM, (**e**) LCHM vs. LCLM, (**f**) HCHM vs. LCLM. The horizontal axis represents the fold change in metabolite expression across different subgroups [log2(FoldChange)], while the vertical axis indicates the significance level of differences [−log10 (*p*-value)]. Each point on the plot represents a metabolite, with red indicating a significant increase, blue indicating a significant decrease, and black indicating no significant difference. Plasma metabolites were visualized on a volcano plot based on their fold change (FC) values, *p*-values, and *q*-values. VIP value indicates the contribution of each variable to the PLS-DA model. Differential metabolites were identified based on VIP values > 1, fold change (FC) > 1.2 or <0.83, *p* < 0.05, and *q* < 0.25.

**Figure 6 metabolites-14-00415-f006:**
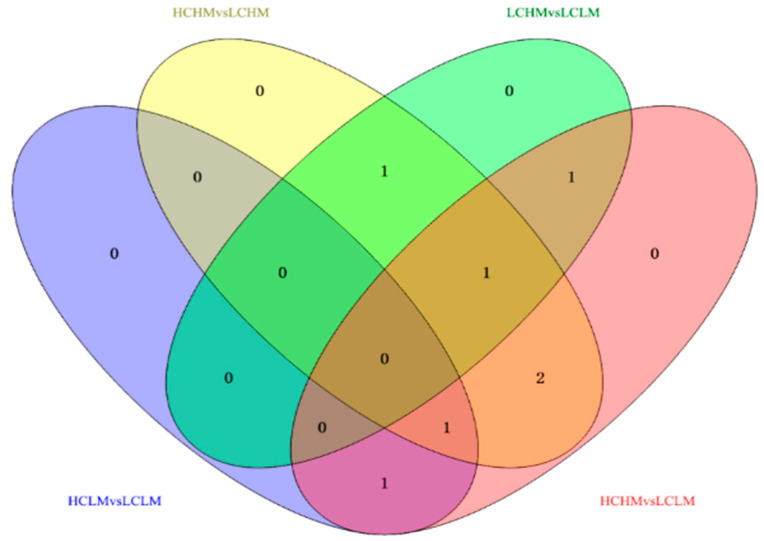
Venn diagram analysis of potential biomarkers in different groups.

**Figure 7 metabolites-14-00415-f007:**
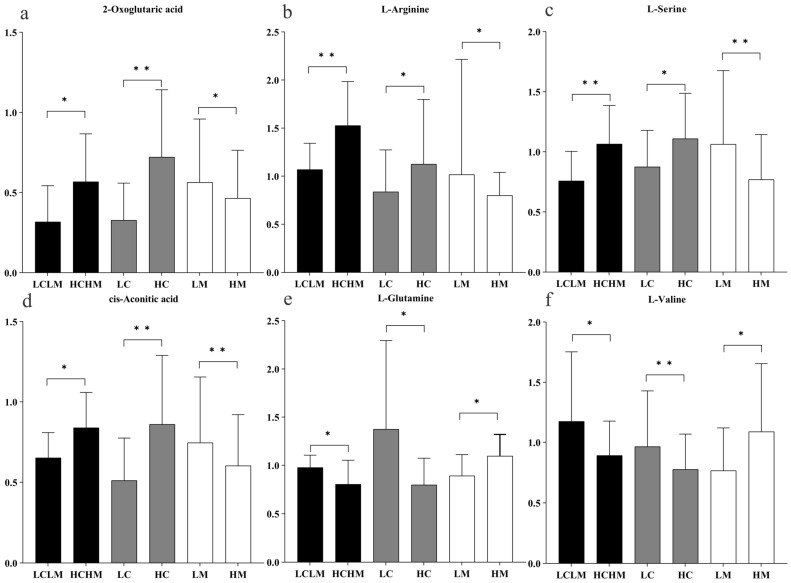
Histogram of trends in common differential metabolites by group. “*” indicating differential metabolite FC > 1.2 or <0.83, *p* < 0.05, and *q* < 0.25; “**” indicating differential metabolite FC > 1.2 or <0.83, *p* < 0.01, and *q* < 0.10; black column: HCHM group vs. LCLM group; grey column: HC group vs. LC group; white column: HM group vs. LM group. (**a**) 2-oxoglutaric acid; (**b**) L-arginine; (**c**) L-serine; (**d**) cis-aconitic acid; (**e**) L-glutamine; (**f**) L-valine.

**Figure 8 metabolites-14-00415-f008:**
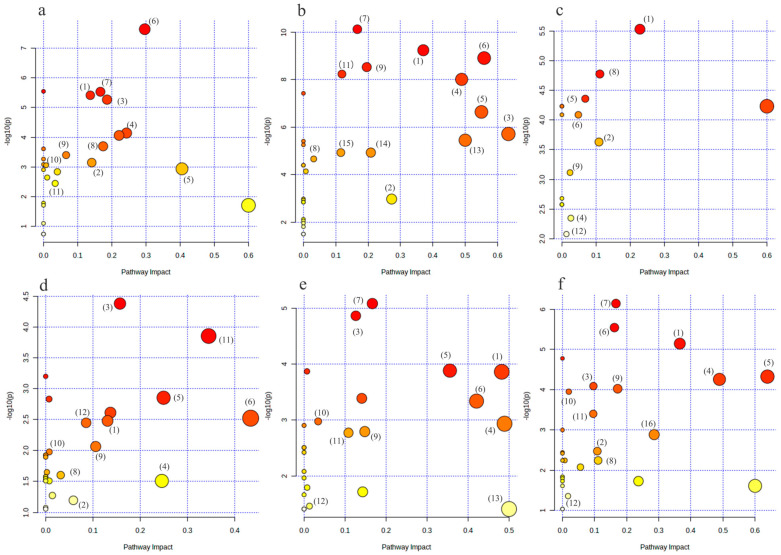
Bubble diagram of metabolic pathways of differential metabolites in each group. (**a**–**f**) represent the metabolic pathway bubble diagrams for each group, (**a**) HC vs. LC group; (**b**) HM vs. LM group; (**c**) HCLM vs. LCLM group; (**d**) HCHM vs. LCHM group; (**e**) LCHM group vs. LCLM group (**f**) HCHM vs. LCLM group (1): arginine biosynthesis; (2): tricarboxylic acid cycle(TCA); (3): cysteine and methionine metabolism; (4): glycine, serine, and threonine metabolism; (5): arginine and proline metabolism; (6): alanine, aspartic acid, and glutamate metabolism; (7): aminotransferase-tRNA biosynthesis; (8): Butyric acid metabolism; (9): glyoxylate and dicarboxylic acid metabolism; (10): purine metabolism; (11): glutathione metabolism; (12): glycerophospholipid metabolism; (13): D-glutamine and D-glutamate metabolism; (14): pyruvate metabolism; (15): glycolysis/glycohydrogenation; (16): taurine and hypotaurine metabolism. The graph’s *X*-axis represents the pathway influence factor, and the *Y*-axis shows the enrichment analysis’s *p*-value. The bigger the circle, the more influential it is; the darker the circle’s color, the lower the *p*-value and the more significant the enrichment.

**Table 1 metabolites-14-00415-t001:** UHPLC-Q-TOF/MS condition.

Name	Conditions of Association
chromatograph	Agilent 1290 Ultra High Performance Liquid Chromatography−6546 Quadrupole-Time of Flight Mass Spectrometer
column	Agilent eclipse plus reversed-phase C18 column (150 mm × 2.1 mm, 1.8 µm particle size)
mobile phase A	0.1% (*v*/*v*) Formic acid–water
mobile phase B	0.1% (*v*/*v*) Formic acid–acetonitrile
Gradient elution	t = 0 min, 25% MPB; t = 10 min, 99% MPB; t = 13 min,99% MPB t = 15 min, 99% MPB; t = 15.1 min, 25% MPB; t = 18 min, 25% MPB
Flow rate	400 µL/min
column temperature	40 °C
Scan range	*m*/*z* 220−1000 Da

**Table 2 metabolites-14-00415-t002:** Basic information of subjects in CRF and MetS groups.

Variables	CRF Group			MS Group	
	LC (*n* = 26)	MC (*n* = 26)	HC (*n* = 27)	LM (*n* = 49)	HM (*n* = 41)
Age (year)	54.19 ± 6.01	53.38 ± 6.33 *	52.85 ± 6.66	53.63 ± 5.98	52.17 ± 6.55
Height (cm)	160.94 ± 6.53	167 ± 8.09	168.63 ± 6.52 *	163.12 ± 6.71	168.63 ± 7.59 ^&^
Weight (kg)	61.42 ± 9.01	67.91 ± 15.63	76.58 ± 24.03 *	61.26 ± 8.40	77.98 ± 20.88 ^&^
BMI (kg/m^2^)	23.63 ± 2.38	24.11 ± 3.89	26.85 ± 8.33	22.96 ± 2.15	27.30 ± 6.82 ^&^
WC (cm)	80.30 ± 6.93	85.98 ± 13.05	88.56 ± 8.96 *	80.07 ± 7.00	90.08 ± 11.28 ^&^
SBP (mmHg)	121.81 ± 17.90	126.69 ± 17.85	130.67 ± 17.70	122.45 ± 17.83	129.51 ± 15.87
DBP (mmHg)	79.42 ± 12.27	78.42 ± 12.81	82.85 ± 12.61	77.88 ± 12.75	82.34 ± 10.83
FPG (mmol/L)	5.39 ± 0.85	5.59 ± 1.43	5.26 ± 0.58	5.17 ± 0.73	5.67 ± 1.14 ^&^
TG (mmol/L)	1.68 ± 1.07	1.65 ± 1.26	2.31 ± 2.00	1.30 ± 0.73	2.61 ± 1.78 ^&^
TC (mmol/L)	5.33 ± 1.00	5.11 ± 1.49	5.01 ± 1.41	5.22 ± 1.25	4.99 ± 1.27 ^&^
HDL-C (mmol/L)	1.21 ± 0.43	1.17 ± 0.33	0.92 ± 2.07 *^#^	1.27 ± 0.37	0.88 ± 0.13 ^&^
LDL-C (mmol/L)	3.57 ± 0.85	3.33 ± 1.27	3.41 ± 1.28	3.50 ± 1.15	3.29 ± 1.04
Hcy (μmmol/L)	11.46 ± 4.35	13.41 ± 4.77	13.18 ± 2.33	11.90 ± 3.80	13.66 ± 4.10 ^&^
VO_2max_	28.90 ± 1.37	35.78 ± 3.41 *	43.18 ± 1.46 *^#^	34.54 ± 6.21	38.04 ± 5.92 ^&^

In the CRF group, “*” for *p* < 0.05 compared with the LC group, and “^#^” for *p* < 0.05 compared with the MC group. In the MS group, “^&^” for *p* < 0.05 compared with the LM group.

**Table 3 metabolites-14-00415-t003:** Basic information of subjects in CRF + MetS groups.

Variables	CRF + MS Group			
	LCLM (*n* = 18)	LCHM (*n* = 8)	HCLM (*n* = 12)	HCHM (*n* = 15)
Age(year)	53.39 ± 6.01	56.00 ± 6.00	54.17 ± 6.71	51.80 ± 6.65
Height(cm)	158.78 ± 4.20	165.81 ± 8.38	168.26 ± 5.15 *	168.93 ± 7.61 *
Weight(kg)	57.98 ± 6.65	69.16 ± 9.16 *	67.72 ± 8.16 *	83.66 ± 29.98 *
BMI(kg/m^2^)	22.97 ± 2.07	25.13 ± 2.47 *	23.84 ± 1.82	29.25 ± 10.61 *
WC(cm)	78.54 ± 5.66	84.25 ± 8.24 *	84.68 ± 7.28 *	91.66 ± 9.18 *^&^
SBP(mmHg)	119.06 ± 18.59	128.00 ± 15.55	121.75 ± 17.92	137.80 ± 14.39 *^&^
DBP(mmHg)	79.17 ± 12.32	80.00 ± 13.00	77.67 ± 12.28	87.00 ± 11.63
FPG(mmol/L)	5.44 ± 0.99	5.28 ± 0.43	4.99 ± 0.30	5.47 ± 0.66 ^&^
TG(mmol/L)	1.44 ± 1.03	2.21 ± 1.02	1.25 ± 0.36 ^#^	3.15 ± 2.37 *^&^
TC(mmol/L)	5.56 ± 0.91	4.81 ± 1.06	4.95 ± 1.73	5.05 ± 1.16
HDL-C(mmol/L)	1.33 ± 0.46	0.93 ± 0.08 *	1.07 ± 0.23	0.82 ± 0.94 *^#&^
LDL-C(mmol/L)	3.74 ± 0.74	3.20 ± 1.01	3.50 ± 1.77	3.34 ± 0.77
Hcy(μmmol/L)	10.74 ± 3.59	13.09 ± 5.67	13.03 ± 3.26	13.30 ± 1.31 *
VO_2max_	28.90 ± 1.38	28.90 ± 1.24	43.30 ± 1.71 *^#^	43.08 ± 1.29 *^#^

In the CRF + MS group, *p* < 0.05 compared with the LCLM group is “*”; *p* < 0.05 compared with the LCHM group is “^#^”; and *p* < 0.05 compared with the HCLM group is “^&^”.

**Table 4 metabolites-14-00415-t004:** PLS-DA cross-validation parameters.

Group	R^2^X	R^2^Y	Q^2^
LC vs. MC vs. HC	0.144	0.481	−0.063
LM vs.HM	0.322	0.967	0.520
LCLM vs. HCLM	0.249	0.981	0.557
LCHM vs. HCHM	0.377	0.992	0.625
LCLM vs. LCHM	0.365	0.992	0.715
HCLM vs. HCHM	0.290	0.946	0.195
LCLM vs. HCHM	0.368	0.997	0.813
LCHM vs. HCLM	0.333	0.998	0.296

**Table 5 metabolites-14-00415-t005:** OPLS-DA cross-validation parameters.

Group	R^2^X	R^2^Y	Q^2^
LC vs. HC	0.259	0.974	0.607
LC vs. MC	0.162	0.930	0.310
MC vs. HC	0.139	0.810	−0.152
LM vs. HM	0.415	0.998	0.520
LCLM vs. HCLM	0.296	0.994	0.548
LCHM vs. HCHM	0.262	0.979	0.520
LCLM vs. LCHM	0.287	0.966	0.613
HCLM vs. HCHM	0.321	0.989	0.028
LCLM vs. HCHM	0.342	0.997	0.651
LCHM vs. HCLM	0.192	0.958	0.459

**Table 6 metabolites-14-00415-t006:** Common differential metabolite FC values for each differential comparison group.

Compounds	CRF Group	MS Group	CRF + MS Group			
	HC vs. LC	HM vs. LM	HCLM vs. LCLM	HCHM vs. LCHM	LCHM vs. LCLM	HCHM vs. LCLM
L-Methionine	0.772 *	1.446 **	-	0.457 *	2.214 *	-
γ-Aminobutyric acid	1.284 *	0.701 *	-	-	-	-
2-Oxoglutaric acid	2.204 **	0.824 *	2.872 **	1.481 *	-	1.784 *
L-Arginine	1.343 *	0.787 *	-	1.251	0.746 **	1.429 **
L-Serine	1.268 *	0.725 **	-	-	0.703 **	1.404 **
cis-Aconitic acid	1.681 **	0.809 **	2.026 **	-	-	1.289 *
L-Glutamine	0.581 *	1.227 *	-	0.783 *	-	0.824 *
L-Valine	0.804 *	1.419 **	-	0.787 *	-	0.759 *

FC values are the multiplicity of differences, “*” indicating differential metabolite FC > 1.2 or <0.83, *p* < 0.05, and *q* < 0.25; “**” indicating differential metabolite FC > 1.2 or <0.83, *p* < 0.01, and *q* < 0.10.

## Data Availability

Data is contained within the article.

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
