# Peer review of "Plasma Metabolomics Study on the Impact of Different CRF Levels on MetS Risk Factors"

_metabolites, 2024, doi:10.3390/metabo14080415_

Round 1

Reviewer 1 Report

Comments and Suggestions for Authors

Dear Author, please recheck and answer the below items:

In the introduction part:

1. How does the study define and categorize different levels of cardiorespiratory fitness (CRF) and ensure these categories are distinct and comparable?

2. What specific criteria were used to select the population sample for the metabolomics analysis, and how representative is this sample of the general population with MetS risk factors?

3. How do the authors address potential confounding variables, such as diet, medication, and lifestyle factors, that could impact both CRF levels and metabolic profiles?

4. Given that previous studies have shown no significant changes in MetS risk factors post-exercise despite increased CRF, how does this study propose to differentiate and measure the impact of CRF on metabolic health beyond traditional risk factors?

In Discussion:

1. How do the authors account for the variation in plasma metabolite levels due to factors other than CRF, such as dietary habits, medication, and genetic predispositions?

2. What are the specific methods and criteria used to identify and validate the eight differential metabolites as biomarkers for distinguishing CRF levels and MetS risk factors?

3. How do the observed differences in plasma methionine levels between the HC vs. LC and HM vs. LM groups reconcile with existing literature on exercise-induced oxidative stress and antioxidant defense mechanisms?

4. Given the conflicting results on GABA levels in different CRF and MetS groups, how do the authors propose to further investigate the role of GABA in modulating obesity, glucose intolerance, and insulin resistance in the context of high CRF levels?

In Limitation and conclusion:

1. How might the metabolic similarities among the LC, MC, and HC groups impact the validity of the study's findings, and what steps could be taken to mitigate this limitation?

2. Given the lower CRF levels of the middle-aged and older participants, how generalizable are the study's conclusions to younger populations or those with higher baseline CRF levels?

3. How does the small sample size in the LCHM group affect the statistical power of the study, and what implications does this have for the reliability of the observed differences in triglyceride levels?

4. What additional research or methodological adjustments are necessary to confirm the role of the six identified metabolites as biomarkers for improving the metabolism of populations with MetS risk factors under high CRF levels? 

Author Response

Comments 1:How does the study define and categorize different levels of cardiorespiratory fitness (CRF) and ensure these categories are distinct and comparable?

Response 1:Thank you for pointing this out.Cardiorespiratory endurance refers to the capacity of the cardiovascular and pulmonary systems to supply oxygen to the body during physical activity.Additionally, the subjects' maximum oxygen uptake was extrapolated using a step test and categorized into three groups based on levels of cardiorespiratory endurance: low, medium, and high. Trichotomization is a widely used statistical method for analyzing data distribution.

Comments 2: What specific criteria were used to select the population sample for the metabolomics analysis, and how representative is this sample of the general population with MetS risk factors?

Response 2: Thank you for pointing this out.According to the Chinese Guidelines for the Prevention and Control of Type 2 Diabetes Mellitus (2020 edition), individuals meeting any one of the five diagnostic criteria (≥1) for metabolic syndrome were included in the study. Individuals meeting one or two diagnostic criteria were also notable as they displayed early signs of developing the syndrome, even though they did not yet meet the complete set of criteria (≥3).This subgroup also requires further study.

Comment 3: How do the authors address potential confounding variables, such as diet, medication, and lifestyle factors, that could impact both CRF levels and metabolic profiles?

Response 3: Thank you for pointing this out.Participants' diets, medications, and lifestyles were assessed before the formal trial (e.g., blood collection, cardiopulmonary endurance test), and eligible subjects were screened based on the inclusion criteria.Moreover, subjects fasted for 10 hours prior to blood collection, and samples were obtained while they remained fasting.To avoid any impact on the test results, the staff ensured that subjects refrained from excessive exercise for three days preceding the cardiorespiratory endurance test.

Comments 4 :Given that previous studies have shown no significant changes in MetS risk factors post-exercise despite increased CRF, how does this study propose to differentiate and measure the impact of CRF on metabolic health beyond traditional risk factors?

Response 4 :Thank you for pointing this out.The study findings indicate that metabolomic analyses can differentiate and quantify changes beyond those related to metabolic syndrome risk factors.High-throughput methods such as nuclear magnetic resonance and mass spectrometry can be used specifically to quantify changes in metabolite composition.In the future, metabolite profiles before and after exercise can be compared, potentially shedding light on the effects of cardiorespiratory endurance on metabolic pathways and metabolites.We focus specifically on metabolites associated with metabolic syndrome, such as blood glucose and lipids

Comments 5 :How do the authors account for the variation in plasma metabolite levels due to factors other than CRF, such as dietary habits, medication, and genetic predispositions?

Response 5 :Thank you for pointing this out.To assess heritability, we gathered and recorded factors such as dietary practices, medication use, and relevant medical histories from study participants. This included information on inherited disorders and medical histories of family members, which could aid in genetic analyses and provide background knowledge for data interpretation.

Comments 6: What are the specific methods and criteria used to identify and validate the eight differential metabolites as biomarkers for distinguishing CRF levels and MetS risk factors?

Response 6: Thank you for pointing this out.Blood samples were analysed using LC-MS metabolomics technology. IsoMS Pro 1.2.16 software was employed for preliminary data processing and analysis, and a three-level metabolite identification method was applied to determine the structure of detected metabolites and collect raw data. Normalised data matrices were formatted and imported into SIMCA-P 14.1 software for multivariate data analysis, including Partial Least Squares Discriminant Analysis (PLS-DA) and Orthogonal Partial Least Squares Discriminant Analysis (OPLS-DA), to compare plasma metabolite differences between groups. Model quality for OPLS-DA was evaluated using cross-validation and a response-ordering test (randomly ranked 200 times) to prevent overfitting. Differential metabolites were selected based on a variable projection importance value (VIP) > 1.0 and non-parametric test criteria (P < 0.05). Specifically, differential metabolites were identified using volcano plot and Wayne diagram analyses, guided by thresholds of VIP > 1, fold change (FC) > 1.2 or < 0.83, P < 0.05, and q < 0.25 (Storey's q-value).

Comments 7: How do the observed differences in plasma methionine levels between the HC vs. LC and HM vs. LM groups reconcile with existing literature on exercise-induced oxidative stress and antioxidant defense mechanisms?

Response 7: Thank you for pointing this out.Future research should investigate the dynamic balance between exercise-induced oxidative stress and the body's antioxidant defence mechanisms. Moderate exercise has been shown to enhance the body's ability to tolerate oxidative stress by boosting antioxidant defence system activity. However, excessive or inappropriate exercise loads may overwhelm the antioxidant system's capacity, resulting in heightened oxidative stress and potential damage to cellular and tissue health.

Comments 8: Given the conflicting results on GABA levels in different CRF and MetS groups, how do the authors propose to further investigate the role of GABA in modulating obesity, glucose intolerance, and insulin resistance in the context of high CRF levels?

Response 8:Thank you for pointing this out. Future metabolomics-targeted studies could include pre- and post-exercise comparisons to quantify GABA levels. This approach would help elucidate the role of GABA in regulating obesity and insulin resistance.

Comments 9:How might the metabolic similarities among the LC, MC, and HC groups impact the validity of the study's findings, and what steps could be taken to mitigate this limitation?

Response 9:Thank you for pointing this out.Utilizing more precise techniques for measuring cardiorespiratory endurance, such as power cycling and cardiopulmonary exercise testing, can enhance accuracy. Increasing the number of participants can mitigate issues related to metabolic state similarity, while direct measurements can provide a more genuine assessment of cardiorespiratory endurance levels.

Comments 10: Given the lower CRF levels of the middle-aged and older participants, how generalizable are the study's conclusions to younger populations or those with higher baseline CRF levels?

Response 10:Thank you for pointing this out.The findings of this study are relevant to both younger and older groups with higher cardiorespiratory fitness (CRF), including participants aged 30 to 60 from the Palmnas study (PMID: 29459697).

Comment 11:How does the small sample size in the LCHM group affect the statistical power of the study, and what implications does this have for the reliability of the observed differences in triglyceride levels?

Response 11:Thank you for pointing this out.Statistical significance is commonly used to evaluate differences between groups. However, in small sample sizes, the p-value may be inflated, potentially leading to nonsignificant results despite the presence of a true difference.

Comments 12:Thank you for pointing this out.What additional research or methodological adjustments are necessary to confirm the role of the six identified metabolites as biomarkers for improving the metabolism of populations with MetS risk factors under high CRF levels?

Response 12 :Thank you for pointing this out.Furthermore, future research should employ additional histological techniques to fully elucidate the biological impact of exercise. Targeted metabolomics tools for quantitative analysis could also be utilized to investigate the significance of potential biomarkers in further studies.

Reviewer 2 Report

Comments and Suggestions for Authors

1. it is better to draw the flowchart for participation and grouping.

2. the exclusion and inclusion criteria need to be precise.

3. the quality of the figures is low, and I could not evaluate them.

4. the author did not mention the analysis method in the statistical analysis.

5. how does the author normalize the samples?

6. the author needs to estimate the sample size.

7. write the analysis method in detail.

8. Write the p-value in the result, write the caption in detail, and explain.

9. what is a home message?

10. write the discussion based on the SOWT.

Author Response

Comment 1:it is better to draw the flowchart for participation and grouping.

Response 1:Thank you for bringing this to our attention; we agree with your comment. We have included the grouping flow chart on page 2 line 115 of the manuscript.

Comments 2:the exclusion and inclusion criteria need to be precise.

Response 2:A total of 90 participants aged 40-65 were recruited from Xiamen City Fujian Province China. Inclusion in the study was based on meeting at least one of the diagnostic criteria for MetS-related risk factors: Central obesity was defined as a waist circumference ≥90 cm for men and ≥85 cm for women;2) fasting triglyceride(TG) level ≥ 1.7 mmol/L or under treatment; 3) high-density lipoprotein cholesterol(HDL-cholesterol) level ≤ 1.04 mmol/L or under treatment; 4) hypertension, defined as a systolic blood pressure (SBP)≥ 130 mm Hg or a diastolic blood pressure(DBP) ≤ 85 mm Hg, or previ-ously diagnosed and treated for hypertension; 5) hyperglycemia, defined as a fasting plasma glucose (FPG)level ≥6.1 mmol/L and/or 2-hour postprandial glucose level ≤ 7.8 mmol/L, or previously diagnosed and treated for T2D.Exclusion criteria encompassed failure to complete the trial or use of vasoactive drugs (e.g., antihypertensive drugs, statins), and dietary supplements during the trial were also excluded. Lines 78-79 and 87-88 of the manuscript contain this information.

Comments 3:the quality of the figures is low, and I could not evaluate them.

Response 3:We appreciate the reviewer's concern regarding the sample size. We chose to include 90 subjects based on findings from studies by Angelika and Marie to ensure adequate power to detect the effects or differences of interest at a statistically significant level (PMID: 24226038, PMID: 29459697). In addition, obtaining blood samples in China is challenging, resulting in relatively small sample sizes.In future studies, we will explore opportunities to increase the sample size through collaborative partnerships or multi-center studies to enhance the robustness of our findings.

Comments 4:the author did not mention the analysis method in the statistical analysis.

Response 4:We very much recognise the comments you have made. We have therefore added the detailed statistical methodology, which is on page 6, lines 189-199 of the text.

Comments 5:how does the author normalize the samples?

Response 5:A total of 97 samples, including 90 experimental samples and 7 quality control (QC) samples, were collected for the liquid analysis.Following collection and export, the data underwent preliminary processing and analysis using IsoMS Pro software. Subsequently, after format conversion and quality assessment, the data were prepared for further analysis.The data analysis parameters included a mass-to-charge ratio (m/z) range of 240 to 1000, a saturation intensity threshold set at 15,000,000, a retention time tolerance of ±9 seconds, and a mass tolerance of ±10 ppm.Only data corresponding to metabolites detected in at least 80% of the samples were retained, aiming to eliminate unstable information.Post-filtering, each dataset was normalised using the total useful signal ratio.

Comments 6: the author needs to estimate the sample size.

Response 6:Thank you for pointing out this issue.We conducted a comprehensive statistical power analysis to determine the minimum sample size required for our study.Additionally, we conducted an extensive literature review on sample sizes utilized in similar studies, considering recommended standards and practical applications in the field

Comments 7:write the analysis method in detail.

Response 7:Thank you for pointing out this issue.We acknowledge the shortcomings in our analysis methods. Consequently, we have included descriptions of subject exclusion and inclusion, a flowchart detailing subject grouping, and additional details on statistical analyses in the study methods section.

Comments 8:Write the p-value in the result, write the caption in detail, and explain.

Response 8:Thank you for your question. We will provide detailed explanations of the result values, including a thorough description of the p-values, as well as ensure clarity in the title of our study.

Comments 9:what is a home message?

Response 9:This study aimed to investigate how variations in CRF levels impact MetS risk factors by conducting a metabolomics analysis on individuals with varying degrees of MetS.

Comments 10:write the discussion based on the SOWT.

Response 10:We totally agree with you on this proposal. We have revised the Discussion section as you suggested.On the page 16 line 431

Reviewer 3 Report

Comments and Suggestions for Authors

The aim of the article "Metabolomic mechanisms of the influence of cardiorespiratory fitness on the risk factors of metabolic syndrome" was to determine through the study how the level of cardiorespiratory fitness affects the risk of metabolic syndrome and what metabolites can serve as biomarkers for these changes. The authors conducted research on a group of 90 patients aged 40-45, analyzing their blood plasma using untargeted metabolomics analysis using the LC-MS method.

According to the reviewer, both the study and the entire work are very interesting. All work is carried out according to the steps of the scientific method. A review of the current literature allows you to present the goal, the methodology is properly selected, the presentation of the results is clear and the conclusions result from the research. The study resulted in the identification of eight common metabolites that could serve as potential biomarkers to distinguish between individuals with different levels of cardiorespiratory fitness and varying degrees of metabolic syndrome risk factors. Increasing cardiorespiratory fitness to some extent may help improve risk factors in metabolic syndrome populations, as high levels of cardiorespiratory fitness protect against high risk of metabolic syndrome. Because metabolites are crucial to understanding cellular processes and disease states, they can also serve as biomarkers for various physiological and pathological states, and thus metabolite profiles can indicate the presence of certain diseases. The study was well planned and conducted on a representative age group, which allows for reliable conclusions. High-quality tools were used for the study, guaranteeing the precision of the study, such as untargeted metabolomic analysis using the LC-M method. Although I recommend printing the work, I would like to present some minor comments or rather suggestions:

The study is limited to the 40-45 age group. It would be worth extending the analysis to a broader age group to assess whether the same relationships exist in younger and older populations. The study focuses on short-term changes in metabolites. Long-term studies could provide more information. The influence of environmental conditions on metabolites in the study population could be described in more detail.

I would like there to be more work that benefits people's health and life.

Author Response

We would like to express our sincere gratitude for your professional review of our article.

The study identified how CRF levels influence metabolic status across varying degrees of MetS through metabolomic analysis, shedding light on CRF's impact on MetS risk via specific metabolic pathways. Changes in these metabolites may correlate with diverse CRF levels and MetS risk, offering novel perspectives for clinical diagnosis and intervention. The findings indicate that higher CRF levels can alleviate the detrimental effects of MetS risk factors, suggesting that enhancing fitness could effectively prevent or treat MetS. Moreover, the identification of six common differential metabolites impacting multiple vital biometabolic pathways enhances our comprehension of metabolic network complexity and proposes potential targets for intervention.

We thank the reviewer for the positive feedback

Round 2

Reviewer 2 Report

Comments and Suggestions for Authors

The author responds to comments.